# Pathological and Molecular Insights into the Early Stage of Multiple System Atrophy

**DOI:** 10.3390/cells14241966

**Published:** 2025-12-10

**Authors:** Makoto T. Tanaka, Yasuo Miki, Tomoya Kon, Fumiaki Mori, Koichi Wakabayashi

**Affiliations:** 1Department of Neuropathology, Biomedical Research Center, Hirosaki University Graduate School of Medicine, Hirosaki 036-8562, Japan; h23gm105@hirosaki-u.ac.jp (M.T.T.); neuropal@hotmail.com (F.M.); koichi@hirosaki-u.ac.jp (K.W.); 2Department of Neurology, Hirosaki University Graduate School of Medicine, Hirosaki 036-8562, Japan; kont0413@hirosaki-u.ac.jp

**Keywords:** multiple system atrophy, preclinical, early-stage, α-synuclein

## Abstract

Recently, studies have increasingly focused on neuropathological and molecular alterations that occur in the early stages of neurodegenerative diseases to understand the primary pathogenesis. This review provides an updated overview of the early pathological and molecular changes in multiple system atrophy (MSA), a neurodegenerative condition characterised by the degeneration of both the striatonigral and olivopontocerebellar systems. In advanced stages of MSA, abnormal α-synuclein accumulates in the cytoplasm and nuclei of oligodendrocytes and neurones. However, in addition to these established pathological hallmarks, previous analyses of preclinical MSA cases have revealed characteristic accumulations of abnormal α-synuclein within and adjacent to the nuclear membrane. Moreover, analyses of cerebrospinal fluid and plasma from patients with MSA within 3 years of disease onset have identified alterations in various proteins and microRNAs linked to neurodegeneration and neuroinflammation. Consistent with these findings, *in vitro* and *in vivo* models of early-stage MSA have demonstrated abnormalities in neurodegeneration, neuroinflammation, and mitochondrial function. Collectively, these observations highlight the primary pathogenesis of early-stage MSA.

## 1. Introduction

To elucidate the pathogenesis of neurodegenerative diseases, it is essential to examine the brains of affected individuals. However, clinicians and neuroscientists frequently encounter complex neuropathological conditions involving not only primary processes but also various secondary processes, such as non-specific inflammation and apoptosis, especially when investigating the brains at advanced stages of the disease. The secondary processes may obscure the primary mechanisms underlying neurodegeneration. Furthermore, clinicians have focused on the early stages of the disease as a therapeutic target because neurones and glial cells are relatively preserved at this stage. Consequently, recent studies have increasingly focused on neuropathological and molecular alterations that occur in the early stages of the disease for diagnosis and treatment. This review aims to update the current knowledge on early pathological and molecular changes in neurodegenerative diseases, with a particular emphasis on multiple system atrophy (MSA).

## 2. MSA

MSA is a rare, sporadic, fatal neurodegenerative disorder clinically characterised by parkinsonism with poor responsiveness to L-DOPA, cerebellar ataxia, and severe autonomic dysfunction [1]. The mean age of onset is 55–60 years, and the average disease duration is approximately 7–9 years after symptom onset [2,3,4]. Clinically, MSA is classified into two subtypes according to the predominant motor features: parkinsonian type (MSA-P) and cerebellar type (MSA-C) [5]. The predominant subtype may change over the course of the disease depending on the prevailing motor symptoms at the time of assessment. MSA-P is more prevalent in European and North American populations, whereas MSA-C predominates in Asian and mestizo populations, probably due to ethnic or genetic factors [3,6,7,8,9]. Although a distinct subtype defined by predominant autonomic dysfunction has not been formally established in the current diagnostic criteria [1], autonomic impairments of varying severity are observed in most patients throughout the disease course. Additionally, the onset, progression, and severity of individual symptoms vary considerably. Recently, several clinicopathological studies revealed that up to 37% of patients with MSA exhibit cognitive impairment in life [10,11].

α-Synuclein (α-Syn) is a 14 kDa neuronal protein that exists as a monomer or multimer at presynaptic terminals, where it regulates synaptic vesicle dynamics and neurotransmitter release [12,13]. However, under pathological conditions, α-Syn undergoes abnormal conformational changes, forming oligomers and subsequently fibrils [14]. Oligomeric α-Syn species are neurotoxic and disrupt synaptic function, whereas fibrillar α-Syn facilitates cell-to-cell pathological propagation [11,15,16,17]. These abnormal α-Syn forms spread along neuronal networks, converting physiological α-Syn into pathological forms. In MSA, abnormal α-Syn accumulates chiefly in the cytoplasm of oligodendrocytes as glial cytoplasmic inclusions (GCIs), the pathological hallmark of the disease (Figure 1a) [18,19]. To a lesser extent, α-Syn also aggregates in the nuclei of oligodendrocytes (glial nuclear inclusions, GNIs; Figure 1b), and in neuronal cytoplasm and nuclei (neuronal cytoplasmic inclusions [NCIs] and neuronal nuclear inclusions [NNIs]; Figure 1c,d) [19]. The GCI density is strongly correlated with the degree of neuronal loss in the striatonigral and olivopontocerebellar systems (Figure 2) [20]. However, in highly advanced stages of the disease, the density of GCIs often declines. Early in the disease course, either the striatonigral or the olivopontocerebellar system is preferentially affected; however, with disease progression, both systems become severely degenerated, often to a comparable degree (Figure 2) [20,21]. Severe putaminal degeneration is typically accompanied by marked involvement of the posterior frontal cortex, including the motor areas. Patients with predominant striatonigral involvement have MSA-P, whereas those with olivopontocerebellar atrophy have MSA-C. Importantly, the variable distribution and extent of α-Syn-positive inclusions (GCIs, GNIs, NCIs, and NNIs) and neuronal loss likely underlie the broad clinical heterogeneity of MSA. Indeed, we previously reported the accumulation of α-Syn oligomers and subsequently increased number of NCIs in the hippocampus as a pathological substrate of cognitive impairment in MSA [11,15,22].

Multiple studies have investigated the genetic and environmental factors that may contribute to the development of MSA. However, the findings of these studies were inconclusive. Polymorphism or variants in *COQ2* and *GBA1* are associated with the occurrence of MSA [23,24,25,26,27,28,29,30,31]. However, these associations have not been confirmed by other cohort studies [32,33,34,35]. Recently, genome-wide association studies and transcriptome-wide association analyses in a European cohort revealed four risk loci and novel susceptibility genes for MSA: *USP38-DT*, *KCTD7*, and *lnc-KCTD7-2* [36]. Conversely, another genome-wide association study on MSA in patients with European ancestry did not identify such genetic risks [37]. Federoff *et al.* conducted a cross-regional genome-wide complex trait analysis, reporting low heritability (2.09–6.65%) in American, British, northern, and southern European cohorts [38]. In contrast, Chelban *et al.* investigated 657 individuals with MSA in the UK, the USA, Germany, and Canada and revealed that individuals carrying more than 250 *FGF14* GAA repeat expansions may have a faster rate of disease progression and diminished life expectancy [39].

## 3. α-Syn Aggregation Processes in GCIs

The answers to the origin of α-Syn in GCIs and the dynamics of its expression are elusive. Miller *et al*. reported an absence of *α-Syn* mRNA in oligodendrocytes from both controls and patients with MSA [40]. *In vivo* studies further demonstrated that exogenous α-Syn can propagate from the neurones to oligodendrocytes [41,42]. In contrast, Asi *et al.* detected *SNCA* transcripts in the oligodendrocytes of healthy human brains using laser capture microdissection and quantitative reverse transcription-polymerase chain reaction [43], which was later corroborated by RNAscope and single-nucleus RNA sequencing [44]. Consistent with these molecular findings, Mori *et al.* showed α-Syn immunoreactivity in glial cytoplasm in the normal human brain [45]. Indeed, *SNCA* transcripts and α-Syn immunoreactivity exist in oligodendrocytes. However, given numerous and widely distributed GCIs that contain abundant abnormal α-Syn, it is unlikely that oligodendrocytes alone constitute the exclusive source of abnormal α-Syn. Interestingly, Mavroeidi *et al.* showed that exogenous administration of human α-Syn fibrils induced endogenous α-Syn aggregation, contributing to the formation of GCI-like inclusions in wild-type mice [46]. More recently, inoculation of the synthetic α-Syn fibril strain 1B, assembled from recombinant human α-Syn, into the brains of both transgenic and wild-type mice induced self-replication of the fibrils and first led to the formation of neuronal inclusions, followed subsequently by the appearance of inclusions in oligodendrocytes [47].

Tubulin polymerisation-promoting protein (p25α) is physiologically expressed in the oligodendroglial nucleus, cytoplasm, and myelin [48,49,50,51], and plays a crucial role in microtubule organisation and stabilisation [52,53,54]. However, in MSA, p25α becomes incorporated into GCIs [50,55] and pathogenically relocalises from the nucleus and myelin sheath to the perinuclear cytoplasm of oligodendrocytes, a process thought to precede α-Syn accumulation [56,57]. Using RNAscope with immunofluorescence, Kon *et al.* demonstrated parallel increases in *p25α* and *SNCA* transcripts within GCIs and reported higher *SNCA* transcript area density in oligodendrocytes containing GCIs or GNIs than in those without the inclusions, suggesting a synergistic contribution to GCI formation [58]. Recently, Sekiya *et al*. utilised proximity ligation assay, which enables the visualisation of protein–protein interaction, and demonstrated the deposition of toxic α-Syn oligomers within the Purkinje cells that do not exhibit any α-Syn-positive inclusions prior to neuronal death [59]. Consistent with the findings, Miki *et al.* examined the brains of patients with MSA and showed that toxic oligomeric α-Syn induces synaptic dysfunction without overt neuronal loss, implicating the role of α-Syn oligomers in early-stage pathology [22]. Collectively, the precise origin of α-Syn, the initial site of its pathological accumulation, and the molecular mechanisms underlying early-stage MSA remain undetermined. Further studies are required to answer these questions.

## 4. Clinical Features in the Early Stage of MSA

In the current consensus diagnostic criteria for MSA, a category of prodromal MSA has been proposed to capture patients in the early stage of the disease. This category comprises three layers: (i) clinical non-motor features, (ii) motor features, and (iii) the absence of exclusion criteria [1]. Among these, non-motor features serve as the entry criterion and include rapid eye movement sleep behaviour disorder (RBD), neurogenic orthostatic hypotension, and urogenital failure. Indeed, approximately 50% of patients with MSA present initially with autonomic symptoms [60], such as cardiovascular autonomic failure, urogenital and sexual dysfunction, and orthostatic hypotension [61]. In addition, RBD may predate the onset of motor or autonomic symptoms in Lewy body diseases (Parkinson’s disease and dementia with Lewy bodies) and in MSA by months or even years. Jung *et al.* assessed the frequency of 13 motor and non-motor signs or symptoms—some regarded as “red flags” for MSA—in 61 patients within three years of disease onset. Dysarthria (98.4%) was the most frequent feature, followed by sexual dysfunction (95.1%), probable RBD (90.2%), constipation (82.0%), snoring (70.5%), dysphagia (68.9%), and stridor (42.6%), in addition to parkinsonism and cerebellar ataxia [62]. These observations support the concept that early-stage MSA is characterised by diverse and often prominent non-motor manifestations. However, neurodegeneration in MSA variably involves the striatonigral, olivopontocerebellar, and autonomic systems, and some patients show involvement of only one or two systems in the early stage. In fact, 9% of patients with MSA do not exhibit autonomic dysfunction at any point during the disease course [63]. Furthermore, other parkinsonian disorders, including Parkinson’s disease and progressive supranuclear palsy, can clinically masquerade as MSA during life, as these patients can also exhibit parkinsonism with severe autonomic dysfunction [64]. Conversely, up to half of patients with early-stage MSA show a favourable response to L-DOPA, reflecting relatively mild neurodegeneration in the putamen [3,6]. Distinguishing such cases from Parkinson’s disease is therefore difficult, particularly in the absence of cerebellar ataxia and/or marked autonomic failure [65]. Compounding this challenge, RBD is not specific to MSA or Lewy body diseases: it can also be caused by progressive supranuclear palsy, narcolepsy, the use of antidepressants, alcohol abuse, and various autoimmune diseases, including limbic encephalitis, anti-IgLON5 antibody–associated disease, multiple sclerosis, and Guillain–Barré syndrome [66,67]. Taken together, these factors underscore that an accurate clinical diagnosis of MSA at an early stage remains challenging when based solely on the constellation of clinical signs and symptoms. The development of reliable biomarkers for early-stage MSA is an urgent, unmet need.

## 5. Pathological and Molecular Alterations in the Early Stage of MSA

In the literature, a number of terms are employed to describe the early stage of MSA, including “preclinical” and “early.” However, these terms do not necessarily refer to the same stage or timing of disease progression. Furthermore, the misconception persists that “minimal change” MSA represents an early stage of the disease. However, this type represents a rare variant of MSA rather than a specific disease stage. In this section, we describe the pathological and molecular changes that characterise each stage and type of MSA.

### 5.1. Preclinical MSA

As of 2025, only six autopsy cases of preclinical MSA have been reported. All these reports primarily focused on neuropathological features, including the extent and distribution of neuronal loss, gliosis, and the occurrence of GCIs and NCIs [68,69,70,71,72], and no studies have specifically investigated molecular alterations in preclinical MSA. Among the six autopsy cases, two showed no neuronal loss in the brain, whereas the remaining four exhibited mild neuronal loss within the striatonigral system. Notably, in all cases, GCIs were consistently and widely distributed throughout both the striatonigral and olivopontocerebellar systems [68,69,70,71,72]. Thus, α-Syn accumulation precedes overt neurodegeneration. In three cases, most of the NCIs were located in the perinuclear region (Figure 3a), whereas NNIs were frequently observed along the inner surface of the nuclear membrane (Figure 3b) [69,71,72]. Furthermore, Wiseman *et al.* reported that α-Syn fibrils within neurones penetrate the nuclear envelope, leading to its disruption and subsequent neuronal death [73]. These findings suggest that the accumulation of α-Syn adjacent to the neuronal nucleus may represent one of the earliest pathogenic events in MSA. In addition, the precise initiation site of α-Syn accumulation within the central nervous system (CNS) remains unclear. Recently, Endo *et al.* developed α-Syn-specific positron emission tomography tracers that may enable *in vivo* detection of early α-Syn pathology [74]. Further studies are warranted to determine the initial site of α-Syn pathology and to elucidate how it ultimately contributes to neurodegeneration in MSA.

### 5.2. Early-Stage MSA

The term “early-stage” MSA has not been clearly defined. In this review, following previous reports [75,76,77], early-stage MSA was defined as the period within 3 years of disease onset. At this stage, varying degrees of neurodegeneration can be observed in the striatonigral, olivopontocerebellar, and autonomic systems. To elucidate the molecular alterations in early-stage MSA, analyses of proteins [78,79,80], cytokines [81], microRNAs (miRNAs) [82,83], and metabolites [84] in the cerebrospinal fluid (CSF) or plasma obtained from patients within 3 years of onset have been performed. CSF levels of parkinsonism-associated deglycase, tau, myelin basic protein (MBP), and neurofilament light chain (NfL), which reflect neurodegeneration in the brain, were significantly higher in patients with MSA than in those with Parkinson’s disease [78,79,80]. In addition, CSF and blood NfL protein levels were reportedly the highest in MSA among several Parkinsonian disorders (Parkinson’s disease, Parkinson’s disease dementia, dementia with Lewy bodies, progressive supranuclear palsy, corticobasal syndrome, essential tremor, and idiopathic rapid eye movement sleep behaviour disorder) [85]. Yamasaki *et al.* revealed that CSF levels of pro-inflammatory cytokines, including interleukin-6, granulocyte-macrophage colony-stimulating factor, and macrophage chemoattractant protein-1, correlate with the disease stage of MSA-C [81]. CSF levels of *miR-24* and *miR-148b*, which are associated with neuronal proliferation and differentiation [86,87], were correlated with the extent of cerebellar ataxia [82]. In contrast, the plasma levels of *miR-671-5p*, implicated in neuroinflammation, oxidative stress, and apoptosis [88,89] differed between MSA-P and MSA-C [83], suggesting that these miRNAs may aid in differentiating early-stage MSA-P and MSA-C. Furthermore, plasma homocysteine, an intermediate amino acid generated during methionine metabolism, is elevated in individuals with early-stage MSA compared with that in healthy controls [84]. Homocysteine exerts neuronal toxicity by increasing oxidative injury in neurones and stimulating N-methyl-D-aspartate receptors [90,91,92]; hence, increased methionine levels may contribute to neuronal degeneration in early-stage MSA. Recently, we isolated and compared extracellular vesicles from the CSF of individuals with Parkinson’s disease and those with early-stage MSA-P. Transcriptomic and proteomic analyses revealed significantly altered levels of transcripts (*RN7SL3*, *RN7SL1*, *miR19B2*, *SYF2P2*, and *S100A7*) and proteins (psoriasin [S100A7] and dystroglycan 1) in patients with MSA-P compared with those with Parkinson’s disease. The mRNA levels of *SYF2P2* and *S100A7*, as well as S100A7 protein levels, were correlated with clinical indices, including I-123-metaiodobenzylguanidine myocardial scintigraphy, asymmetry index on dopamine transporter scans, and Mini-Mental State Examination scores. Although the roles of these transcripts and proteins in the early pathogenesis of MSA remain unclear, these transcriptomic and proteomic signatures may help identify early-stage MSA-P [93]. Collectively, these findings indicate characteristic alterations in protein, cytokine, miRNA, and metabolite levels during the early stages of MSA. The alterations of transcripts and proteins in early-stage MSA are summarised in Table 1.

### 5.3. “Minimal Change” MSA

Typically, in MSA, varying degrees of neuronal loss are observed within the striatonigral and olivopontocerebellar systems [94,95], accompanied by the widespread occurrence of GCIs and NCIs throughout the brain [96,97]. In contrast, “minimal change” MSA represents an atypical pathological variant characterised by neuronal loss restricted to specific regions with an otherwise typical distribution of GCIs [94,98,99,100,101,102]. To date, nine autopsy cases with minimal changes in MSA have been reported [98,99,100,101,102]. The initially described affected regions were the substantia nigra and locus coeruleus [94,99,101,102]. Compared with typical MSA, patients with “minimal change” MSA showed a greater burden of NCIs, but not GCIs, in the caudate and substantia nigra [99]. However, subsequent reports documented cases in which neuronal loss was restricted to the olivopontocerebellar system [100] or limbic system [98]. These cases were likewise classified as “minimal change” MSA. Ling *et al.* described the clinical and pathological features of six cases of “minimal change” MSA, noting that individuals with this variant had a shorter disease duration and more rapid disease progression [99]. Thus, the term “minimal change” MSA may encompass a broader pathological spectrum ranging from a rare pathological variant to a potential early stage of the disease.

## 6. *In Vitro* and *In Vivo* Models of Early-Stage MSA

It is difficult to infer the pathological dynamics of early MSA solely from autopsied brain tissue. Therefore, various *in vitro* and *in vivo* models have been developed to recapitulate the key aspects of early pathogenic processes in MSA. In this section, we describe in detail the principal features and advantages of the representative models.

### 6.1. Induced Pluripotent Stem (iPS) Cell Models

#### 6.1.1. Oligodendrocytes

In 2015, Djelloul *et al.* first established iPS cells derived from patients with MSA [103]. They generated iPS cells from fibroblasts of a healthy control and individuals with MSA-C and MSA-P, and differentiated these iPS cells into oligodendrocyte lineage cells to investigate α-Syn expression. Consistent with studies using the brains of patients with MSA [43,58], *SNCA* transcripts were detected in MSA-associated iPS cell-derived oligodendrocytes. α-Syn was detected in O4-positive-oligodendrocytes at 60 days *in vitro* (DIV) and localised to the perinuclear region from 65 DIV onwards. Although α-Syn expression persisted in oligodendrocytes until 110 DIV, its levels decreased in O4-positive oligodendrocytes over time.

Azevedo *et al.* performed transcriptome analysis of oligodendrocyte lineage cells derived from fibroblast iPS cells of patients with MSA-P and MSA-C and healthy controls [104]. Transcriptomic profiling of O4-positive oligodendrocytes at 130 DIV revealed positive enrichment of antigen presentation and processing pathways and negative enrichment of lipid metabolism in MSA-P and MSA-C cells. In addition, distinct biological processes were differentially enriched between MSA-P and MSA-C O4-positive oligodendrocytes; peptide biosynthesis and translation were predominant in MSA-P cells, whereas intracellular protein transport and protein targeting were enriched in MSA-C cells. These findings suggest that distinct molecular pathways underlie the early pathogenic mechanisms of different MSA subtypes.

#### 6.1.2. Neural Progenitor Cells

Using iPS cells derived from fibroblasts obtained from healthy controls and two patients with MSA-P, Herrera-Vaquero *et al.* differentiated iPS cells into neural progenitor cells to investigate early pathogenic mechanisms in MSA neurones [105]. In the study, they examined α-Syn expression levels, mitochondrial alterations, and resistance to exogenous oxidative stress in neural progenitor cells from 7 DIV onwards. While *SNCA* mRNA expression levels were unchanged between the two groups, α-Syn protein was aberrantly translocated into the nucleus in neural progenitor cells derived from patients with MSA-P. In contrast to the intermediate mitochondrial morphology observed in control neural progenitor cells, the mitochondria appeared to be tubulated in cells derived from individuals with MSA-P. Additionally, compared with control neural progenitor cells, MSA neural progenitor cells produced more reactive oxygen species and underwent apoptosis. However, no difference was observed in the mitochondrial area or mitochondrial respiratory function between progenitor cells from individuals with MSA-P and controls.

#### 6.1.3. Dopaminergic Neurones

Compagnoni *et al.* generated dopaminergic neurones from iPS cells derived from the fibroblasts of healthy controls and patients with MSA-P or MSA-C [106]. In their study, the levels of synapsin I (a synaptic marker) and tau protein (a neuritic marker) were significantly reduced in MSA dopaminergic neurones compared with the controls at 70 DIV. In contrast, α-Syn expression levels did not differ between MSA and control neurones at either 35 or 70 DIV. Furthermore, autophagic flux, as indicated by LC3-II levels, and lysosomal enzyme activities (α- and β-mannosidase) were decreased in MSA-affected neurones. In contrast to the findings reported by Herrera-Vaquero *et al.* [105], decreased mitochondrial respiratory chain activity was observed in complexes II and III, accompanied by compensatory increases in the expression of these complexes. The expression levels of coenzyme Q (CoQ) 10 biosynthetic enzymes (all-trans-polyprenyl-diphosphate synthase 1 and 2, ubiquinone biosynthesis protein COQ4 homologue, and atypical kinase COQ8A) are also upregulated in MSA-associated dopaminergic neurones. Interestingly, autophagic and mitochondrial abnormalities differed between MSA subtypes and were more pronounced in MSA-P than in MSA-C-associated dopaminergic neurones.

#### 6.1.4. Striatal GABAergic Medium-Sized Spiny Neurones

Henkel *et al.* investigated how α-Syn accumulation leads to neurodegeneration in medium-sized striatal GABAergic spiny neurones generated from fibroblast-derived iPS cells [107]. Although *SNCA* expression levels did not differ between MSA-P and control cells at 70 DIV, the fluorescence intensity of α-Syn in the cytoplasm and neurites was increased in MSA-P cells compared with controls. In contrast, electrophysiological and Ca^2+^ imaging analyses have revealed significantly reduced excitability of medium-sized striatal GABAergic spiny neurones in MSA. Moreover, MSA-derived neurones released higher levels of α-Syn into the culture supernatant than control neurones, suggesting the acceleration of cell-to-cell propagation of α-Syn in MSA. Recently, Smandzich *et al.* performed a proteomic analysis using striatal GABAergic medium-sized spiny neurones derived from fibroblast iPS cells at 70 DIV [108]. They identified significant alterations in 151 proteins in patients with MSA-P compared with the controls. Among these, a protein involved in CoQ10 biosynthesis (2-methoxy-6-polyprenyl-1,4-benzoquinol methylase) was up-regulated, whereas Ca^2+^ channel-associated proteins (voltage-dependent calcium channel subunit alpha-2/delta-1 and voltage-gated potassium channel subunit Kv7.2) were down-regulated. Upstream regulator analysis further identified the molecular chaperone MKKS, proteins related to translational regulation (Histone H1.1 and lysine-specific demethylase 5A), and the cellular tumour antigen p53, a transcription factor mediating apoptosis, oxidative stress, and DNA damage responses, as potential upstream regulators.

Multiple analyses of oligodendrocytes, neural progenitor cells, dopaminergic neurones, and striatal GABAergic medium spiny neurones have demonstrated the involvement of mitochondrial dysfunction, apoptosis, oxidative stress, and DNA damage responses in the pathogenesis of early-stage MSA. Regardless of the cell type examined, morphological and functional abnormalities of the mitochondria appear to be common features of early-stage MSA. All *in vitro* models of MSA are summarised in Table 2.

### 6.2. Animal Models of MSA

#### 6.2.1. Adult-Onset Proteolipid Protein (PLP) α-Syn Mouse Model of MSA

Multiple transgenic animal models of MSA have been developed to elucidate its pathogenesis. However, a major limitation of the animal models that overexpress human α-Syn in oligodendrocytes is that disease-related changes occur from the time of fertilisation. To overcome the limitations of conventional transgenic animal models, Tanji *et al.* established an adult-onset, human α-Syn-inducible mouse model of MSA under the control of the PLP promoter using the Cre recombinase–loxP system [109]. Along with the formation of toxic α-Syn oligomers, human α-Syn-positive GCI-like structures were observed, followed by the appearance of NCI-like structures. These GCI-like structures became proteinase K-resistant 50 weeks after human α-Syn induction. Consistent with the findings obtained from preclinical MSA cases [69,71,72], human α-Syn was deposited in the perinuclear regions of neurones in the brainstem and putamen, although human α-Syn did not form fibrils in this model. Despite these similarities, this model also exhibits phenotypic features that are not typically observed in human MSA. Cognitive impairment occurs in approximately 37% of patients with MSA, generally at advanced disease stages, in conjunction with the classical triad of Parkinsonism, cerebellar ataxia, and autonomic dysfunction [10]. In contrast, this mouse model displays memory impairment as early as 4 weeks after human α-Syn induction, followed by motor deficits 1 year later [15,109,110,111]. Recently, we performed a transcriptome analysis using the brains of this model [22]. This revealed altered expression levels of transcripts encoding 10 vesicular transport and synaptic proteins, including synaptotagmin 13. Importantly, neuropathological and biochemical analyses of human MSA brains confirmed that the dysregulation of synaptotagmin 13 was caused by the accumulation of toxic α-Syn oligomers at synapses [22]. Taken together, this MSA mouse model recapitulates some key aspects of the pathogenesis of early-stage MSA.

#### 6.2.2. Adeno-Associated Virus (AAV) Models

Adult mouse, rat, and non-human primate models of MSA using viral vectors have been developed [112,113,114,115,116]. In these models, AAV vectors were injected into the striatum of these models to induce human α-Syn in targeted cells and regions, thereby recapitulating the early stages of MSA.

Williams *et al.* injected an oligotrophic AAV vector (Olig001-SYN) into the striatum of C57BL/6 mice to investigate the role of neuroinflammation in early MSA pathogenesis. Oligodendrocyte transduction by this vector in the dorsal striatum and corpus callosum led to demyelination with marked neuroinflammation [112]. In the affected regions, resident microglia display increased major histocompatibility complex class II expression along with the infiltration of proinflammatory monocytes and CD4-positive T cells into the brain [112]. In a subsequent study, the same research group demonstrated that interferon-γ secreted by CD4-positive T cells drives α-Syn–mediated neuroinflammation and neurodegeneration [113]. Given the increased numbers of HLA-DR-positive microglia in the putamen and substantia nigra of postmortem brains with MSA, these findings suggest that T-cell priming and infiltration into the CNS may contribute to the pathogenesis of early-stage MSA.

Two types of rat models of MSA have been developed using the chimeric AAV1/2 vector carrying the human α-Syn sequence under the myelin basic protein MBP promoter [115], or the Olig001 vector encoding human α-Syn [116]. In the first model, GCI-like structures were observed in the striatum, substantia nigra, and corpus callosum [115]. Neuronal loss occurred in the substantia nigra 3 months after AAV injection and in the striatum 6 months after injection. Behaviourally, this rat model developed motor deficits that were unresponsive to L-DOPA for 2 months, and exhibited sensorimotor impairments for 6 months. In the second model, GCI-like structures were similarly observed in the striatum, substantia nigra, and corpus callosum, as well as in the globus pallidus and thalamus [116]. These α-Syn-positive structures were proteinase K-resistant [115,116] and colocalised with p25α [116], resembling the pathological features observed in human MSA.

Non-human primate models have been established to further recapitulate the pathogenic processes of human MSA. AAV vectors have been injected into the putamen and caudate nucleus of rhesus macaques (*Macaca mulatta*) [114] and into the putamen of cynomolgus macaques (*Macaca fascicularis*) [116]. In the rhesus macaque MSA model, aggregated α-Syn was observed in oligodendrocytes within the caudate nucleus, putamen, and corpus callosum [114]. Proteinase K-resistant GCI-like structures were formed in the striatum. This model also exhibited demyelination in the corpus callosum and striatum 3 months after the induction of human α-Syn expression. Consistent with the regional distribution of α-Syn, microglia were activated in the affected regions [114]. Activated microglia have also been observed in the substantia nigra. In a cynomolgus macaque model, thioflavin S-positive GCI-like structures have been observed in the putamen [116]. Six months after the AAV injection, this model exhibited demyelination and neuronal loss in the putamen, accompanied by neuronal degeneration in the substantia nigra. Similarly to the rhesus macaque model, microglia were activated in the putamen along with T cell infiltration. All genetically modified and AAV animal models of MSA are summarised in Table 3.

#### 6.2.3. Inoculation Models

Several species of abnormal α-Syn fibrils have been injected into the brains of transgenic and wild-type mice to investigate how pathological α-Syn propagates *in vivo*. Previously, *A53T* transgenic mice inoculated with brain homogenates from patients with MSA exhibited significantly increased levels of insoluble phosphorylated α-Syn and α-Syn oligomers compared with non-inoculated controls [117,118]. However, contrary to expectations, α-Syn aggregates developed predominantly in neuronal cell bodies and neurites, rather than in oligodendrocytes [117,118]. Similarly, inoculation of MSA brain homogenates into Tg (SNCA)1Nbm/J mice resulted in human α-Syn–positive aggregates largely restricted to neurons [119]. Moreover, inoculation of MSA or PD brain homogenates into wild-type mice did not alter the proportion of α-Syn localised to oligodendrocytes. In contrast, two recent studies from Université de Bordeaux reported that a synthetic, thioflavin T-negative α-Syn fibril strain (1B fibril), assembled from human recombinant α-Syn, self-replicated and induced MSA-like pathology in mice [47,120]. Interestingly, 1B fibrils caused phosphorylated α-Syn-positive structures in the nucleus of primary mouse neurons [121]. Injection of 1B fibrils into the striatum of wild-type mice produced phosphorylated α-Syn–positive neurites spreading along neuronal connectivity [120]. These neurites became fragmented, contorted, and curled six months after inoculation, and importantly, both GCI- and GNI-like structures then emerged in parallel with these structural alterations [120]. A structural analogy between the 1B fibril fold and the human MSA fibril fold was identified in one of the patients with MSA reported by Goedert *et al.* [47,122]. The 1B fibrils closed thioflavin T-binding pockets [47], suggesting that such closed pockets may directly contribute to the formation of MSA-like pathology. All *in vivo* inoculation models of MSA are summarised in Table 4.

### 6.3. Other Models

Three major transgenic mouse models of MSA have been established, in which human α-Syn is overexpressed under the control of different oligodendroglial promoters, including PLP [123], MBP [124], and 2′,3′-cyclic nucleotide 3′-phosphodiesterase, respectively [125]. These models exhibit key pathological and clinical features of MSA, such as the formation of GCI-like structures, motor deficits, and autonomic dysfunction [123,124,125,126,127,128,129,130,131,132,133]. However, α-Syn overexpression occurs from the stage of fertilisation, limiting their suitability for investigating the pathogenic processes underlying early-stage MSA.

## 7. Conclusions

In the present review, we comprehensively summarised the current understanding of early-stage MSA. Notably, several shared features, such as mitochondrial dysfunction and specific neuroinflammation, consistently emerged in both the *in vitro* and *in vivo* MSA models. However, critical questions remain regarding whether these features reflect genuine pathogenic processes or are influenced by methodological differences and species-specific factors. In fact, both *in vitro* and *in vivo* models are designed to replicate some of the key features of early-stage MSA, based on the hypothesised sources and propagation pathways of abnormal α-Syn. These models do not fully replicate the complexity of human disease dynamics, such as onset timing and species-specific factors. Further studies using postmortem brains from patients with preclinical or early-stage MSA are warranted to clarify the upstream mechanisms driving the pathogenesis of MSA.

## Figures and Tables

**Figure 1 cells-14-01966-f001:**
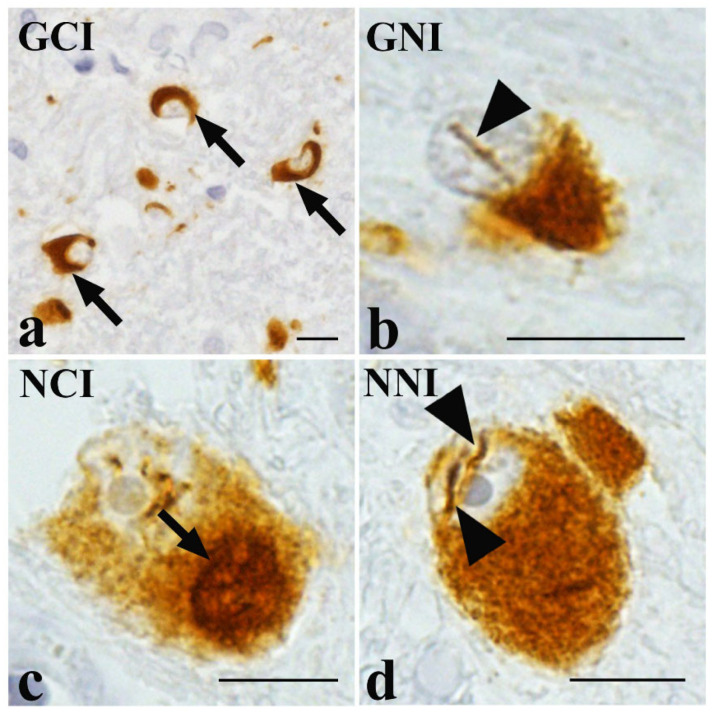
Typical pathological features in multiple system atrophy (MSA). (**a**) α-Synuclein-positive glial cytoplasmic inclusions (GCIs, arrows). (**b**–**d**) α-Synuclein-positive glial nuclear inclusions (GNIs, arrowhead) (**b**), neuronal cytoplasmic inclusions (NCIs, arrow) (**c**), and neuronal nuclear inclusions (NNIs, arrowheads) (**d**). Phosphorylated α-synuclein immunostaining (pSyn#64; FUJIFILM Wako, Osaka, Japan; 1:5000). Scale bars = 10 μm. This figure is from the corresponding author’s laboratory and illustrates the pathological concept of typical MSA.

**Figure 2 cells-14-01966-f002:**
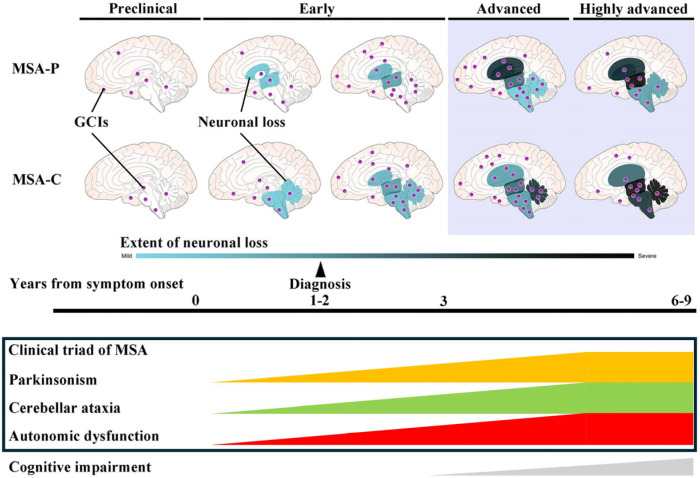
Hypothetical schematic view of disease progression in multiple system atrophy (MSA). Distribution of neurodegeneration and glial cytoplasmic inclusions (GCIs) in the brains of patients with MSA-parkinsonian (MSA-P) and MSA-cerebellar (MSA-C) subtypes. In the early stages of MSA, the striatonigral or olivopontocerebellar system is preferentially affected. As the disease progresses, both systems become markedly involved, typically to a comparable degree in advanced stages. Notably, GCIs are widely distributed beyond these systems, even from the early stages. However, in highly advanced stages of the disease, the density of GCIs often declines.

**Figure 3 cells-14-01966-f003:**
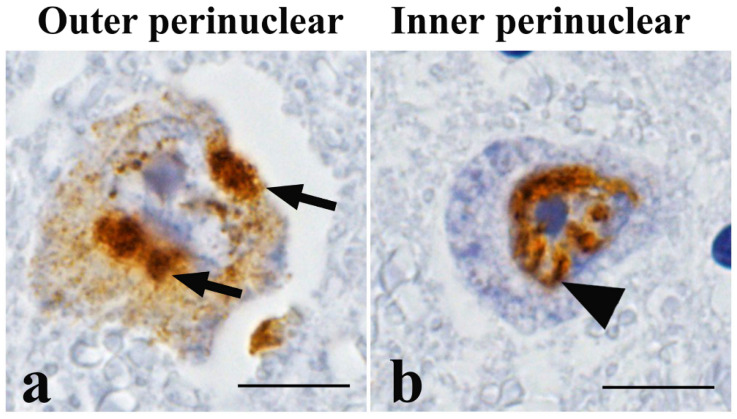
Neuronal cytoplasmic inclusions in preclinical multiple system atrophy (MSA). In the early stages of MSA, α-synuclein accumulates outside ((**a**), arrows) and inside ((**b**), arrowhead) the neuronal nucleus. Immunostaining for phosphorylated α-synuclein. Scale bars = 10 μm. This figure is from the corresponding author’s laboratory and illustrates the pathological concept of early-stage MSA.

**Table 1 cells-14-01966-t001:** Alterations of transcripts and proteins in early-stage MSA.

Disease	N	Sex (Male:Female)	Age, Mean (Years)	Disease Duration, Mean (Years)	Sample	Alterations in Early-Stage MSA	Reference
MSA	23	14:9	60.5	2.8	CSF	**Proteins** DJ-1 levels ↑ (MSA vs. Cont, MSA vs. PD). Tau levels ↑ (MSA vs. PD). **ROC analysis** DJ-1: AUC = 0.84, sensitivity = 78%, specificity = 78% (MSA vs. PD). DJ-1, Tau, and phosphorylated tau (181): AUC = 0.92, sensitivity = 82%, specificity = 81% (MSA vs. PD).	[78]
PD	43	29:14	58.9	3.12			
Non-neurological cont	30	22:13	57.0	NA			
MSA	21	11:10	69.2	2.9	CSF	**Proteins** NfL levels ↑ (MSA vs. Cont). Tau levels ↑ (MSA vs. Cont). **ROC analysis** NfL: AUC = 0.7 (MSA vs. PD). NfL: AUC = 0.7 (MSA vs. PSP). Tau: AUC = 0.69 (MSA vs. PD).	[79]
PD	36	18:18	71.3	2.2			
PSP	20	10:10	73.2	2			
Non-neurological cont	30	16:14	69.6	NA			
MSA	22	15:7	60.7	2.83	CSF	**Proteins** MBP levels ↑ (MSA vs. Cont, MSA vs. PD). **ROC analysis** MBP: AUC = 0.781 (MSA vs. PD).	[80]
PD	55	38:17	57.1	2.85			
Non-neurological cont	118	66:52	55.9	NA			
MSA-C	20	10:10	61.3	2.1	CSF	**Proteins** Positive correlation of IL6 levels with the grade of hot cross ban sign. Positive correlation of IL6 levels with the length of the vermis, pontine base, and medulla oblongata. Positive correlation of GM-CSF levels with the length of the medulla oblongata. Positive correlation of MCP-1 levels with disease durations.	[81]
MSA	17	13:4	62.5	2.14	CSF	**Transcripts** Negative correlation of *MiR-24* levels with ICARS score. Negative correlation of *MiR-148b* with ICARS score.	[82]
MSA (discovery cohort)	13	6:7	62.6	2.67	plasma	**Transcripts** *MiR-661-5p* levels ↓ (MSA vs. Cont). *MiR-661-5p* levels ↓ (MSA-P vs. MSA-C, MSA-P vs. Cont).	[83]
Healthy cont (discovery cohort)	6	3:3	60.7	NA			
MSA-P (validation cohort)	30	13:17	68.1	3.01			
MSA-C (validation cohort)	31	15:16	60.7	3.09			
Healthy cont (validation cohort)	28	15:13	63.2	NA			
MSA	161	82:79	58	2.35	plasma	**Metabolite** Homocysteine levels ↑ (MSA vs. Cont).	[84]
Healthy cont	161	78:83	57.3	NA			
MSA-P	16	4:12	66.4	2.4	CSF	**Transcripts** *RN7SL3* levels ↑ (MSA vs. ALS, MSA vs. PD). *RN7SL1* levels ↑ (MSA vs. ALS, MSA vs. PD). MiR-19B2 levels ↑ (MSA vs. ALS, MSA vs. PD). *SYF2P* levels ↑ (MSA vs. ALS, MSA vs. PD). *S1007A* levels ↓ (MSA vs. ALS, MSA vs. PD). **Proteins** Psoriasin (S1007A) levels ↑ (MSA vs. ALS, MSA vs. PD). DAG1 levels ↓ (MSA vs. ALS, MSA vs. PD).	[93]
PD	16	9:7	71.8	5.4			
ALS	16	12:4	68.3	1.06			

MSA: Multiple system atrophy, Cont: control, PD: Parkinson’s disease, MSA-P: MSA-parkinsonian type, MSA-C: MSA-celebellar type, ALS: Amyotrophic lateral sclerosis, CSF: cerebrospinal fluid, DJ-1: Parkinsonism-associated deglycase, vs.: versus, ROC: Receiver operating characteristic, AUC: Area under the curve, NfL: Neurofilament light chain, MBP: Myelin basic protein, IL6: Interleukin-6, GM-CSF: Granulocyte-macrophage colony-stimulating factor, MCP-1: Macrophage chemoattractant protein-1, miR: microRNA, MRI: magnetic resonance imaging, ICARS: International cooperative ataxia rating scale, DAG1: Dystroglycan 1.

**Table 2 cells-14-01966-t002:** Characteristics and alterations of iPS cell models.

Sample	Cell Type	Characteristics/Alterations	Studies
MSA-P	Oligodendrocytes	**Transcript** Detection of *SNCA* transcripts. Positive enrichment of antigen presentation and processing pathway in MSA-P vs. HC, and MSA-C vs. HC. Negative enrichment of lipid metabolism in MSA-P vs. HC, and MSA-C vs. HC Enrichment of peptide biosynthetic process and translation in MSA-P vs. HC. Enrichment of Intracellular protein transport and protein targeting in MSA-C vs. HC. **Protein** Localisation of α-Syn to the perinuclear region. Decreased expression levels of α-Syn.	[103,104]
MSA-C			
HC			
MSA-P	Neural progenitor cells	**Protein** Translocation of α-Syn into the nucleus. **Mitochondria** Tubulation of mitochondria in MSA-P iPS cells. **Resistance against exogenous oxidative stress** Reactive oxygen species production ↑ (MSA-P vs. HC). Apoptosis ↑ (MSA-P vs. HC).	[105]
HC			
MSA-P	Dopaminergic neuron	**Protein (MSA vs. HC)** Synapsin I levels ↓. Tau levels ↓. **Mitochondria (MSA vs. HC)** Respiratory chain activity (complex II and II + III) ↓. Complex II protein levels ↑. Complex III protein levels ↑. PDSS1 protein levels ↑. PDSS2 protein levels ↑. COQ4 protein levels ↑. COQ8A protein levels ↑. **Autophagy (MSA vs. HC)** Autophagy flux indicated by LC3-II ↓. α-mannosidase activity ↓. β-mannosidase activity ↓.	[106]
MSA-C			
HC			
MSA-P	Striatal GABAergic medium-sized spiny neurons	**Protein** α-Syn in the cytoplasm and neurites ↑. Extracellular α-Syn ↑. COQ5 levels ↑. CACNA2D1 levels ↓. KCNQ2 levels ↓. **Electrophysiological character** Frequency of Ca^2+^ transients ↓. Proportion of cells that exhibit spontaneous action potential firing ↓. Frequency of Action potential ↓. Frequency and amplitude of miniature postsynaptic currents ↓ **Upstream regulator analysis (protein)** Molecular chaperone MKKS. Histone H1.1. KDM5A. TP53.	[107,108]
HC			

iPS cell: induced pluripotent stem cell, MSA: multiple system atrophy, MSA-P: MSA-parkinsonian, MSA-C: MSA-celebellar, HC: Healthy control, vs.: versus, α-Syn: α-Synuclein, PDSS: All-trans-polyprenyl-diphosphate synthase, COQ4: ubiquinone biosynthesis protein COQ4 homologue, COQ8A: atypical kinase COQ8A, LC3-II: Microtubule-associated protein 1 light chain 3 phosphatidylethanolamine conjugate, COQ5: 2-methoxy-6-polyprenyl-1,4-benzoquinol methylase, CACNA2D1: voltage-dependent calcium channel subunit alpha-2/delta-1, KCNQ2: voltage-gated potassium channel subunit Kv7.2, KDM5A: lysine- specific demethylase 5A, TP53: cellular tumour antigen p53.

**Table 3 cells-14-01966-t003:** Characteristics and alterations of genetically modified and AAV-injected animal models.

Sample	Animal Species	Methods for α-Syn Expression	Characteristics/Alterations	Studies
Human α-Syn expressed Tg mouse	Mouse	Proteolipid protein promoter with cre recombinase−loxP system	**Pathology** GCI-like inclusions. Proteinase K-resistant GCI-like inclusions. NCI-like inclusions. Deposition of human α-Syn in the perinuclear region. No formation of α-Syn fibrils. **Symptom** Cognitive impairment. Motor deficits. **Transcript** *SYT13* levels ↑.	[15,109,110,111]
Tg mouse without α-Syn expression				
AAV-α-Syn mouse	Mouse	Oligotrophic AAV vector encoding human α-Syn	**Pathology** GCI-like inclusions. Demyelination. **Neuroinflammation** MHCII protein levels ↑. Infiltration of proinflammatory monocytes and CD4-positive T cells ↑. Secretion of interferon-γ by CD4-positive T cells.	[112,113]
AAV-GFP mouse				
AAV-α-Syn rat	Rat	Chimeric AAV1/2 vector carrying human α-Syn under myelin basic protein promoter	**Pathology** GCI-like inclusions. Proteinase K-resistant GCI-like inclusions. Neuronal loss. **Symptom** L-DOPA unresponsive motor deficits. Sensorimotor impairment.	[115]
AAV-GFP rat				
AAV-α-Syn rat	Rat	Oligotrophic AAV vector encoding human α-Syn	**Pathology** GCI-like inclusions. Proteinase K-resistant GCI-like inclusions. Demyelination. Neuronal loss. **Protein** Colocalisation of α-Syn with p25α	[116]
AAV-GFP rat				
AAV-α-Syn monkey	Monkey (rhesus macaques)	Oligotrophic AAV vector encoding human α-Syn	**Pathology** GCI-like inclusions. Proteinase K-resistant GCI-like inclusions. Demyelination. **Neuroinflammation** Microglial activation ↑.	[114]
AAV-GFP monkey				
AAV-α-Syn monkey	Monkey (cynomolgus macaques)	Oligotrophic AAV vector encoding human α-Syn	**Pathology** GCI-like inclusions. Thioflavin S-positive GCI-like inclusions. Demyelination. Neuronal loss. **Neuroinflammation** Microglial activation ↑. Infiltration of CD3-positive T cell ↑.	[116]
AAV-GFP monkey				

α-Syn: α-Synuclein, AAV: Adeno-associated virus, Tg: transgenic, GFP: Green fluorescent protein, MSA: multiple system atrophy, GCI: Glial cytoplasmic inclusion, NCI: Neuronal cytoplasmic inclusion, MHCII: Major histocompatibility complex II, CD: Cluster of differentiation, p25α: Tubulin polymerization promoting protein.

**Table 4 cells-14-01966-t004:** Characteristics and alterations of inoculation models.

Sample	Animal Species	Inoculated α-Syn	Characteristics/Alterations	Studies
A53T Tg mouse with inoculation	Mouse	MSA brain homogenates	**Pathology** Phosphorylated α-Syn-positive neurons. Gliosis. **Protein** Insoluble phosphorylated α-Syn ↑. Phosphorylated α-Syn positive oligomers ↑. **Neuroinflammation** Microglial activation ↑.	[117,118]
A53T Tg mouse without inoculation				
Tg(SNCA)1Nbm/J mouse with inoculation	Mouse	MSA brain homogenates	**Pathology** Phosphorylated α-Syn-positive neurons.	[119]
Tg(SNCA)1Nbm/J mouse without inoculation				
Wild-type mouse with inoculation	Mouse	Synthetic, thioflavin T-negative α-Syn fibril strain (1B fibril)	**Pathology** NCI-like inclusions. NNI-like inclusions. Phosphorylated α-Syn-positive neurites. GCI-like inclusions. GNI-like inclusions. **Protein** Internalisation of abnormally morphic α-Syn into oligodendrocytes. Subsequent formation of α-Syn positive inclusions in oligodendrocytes.	[47,120]
Wild-type mouse without inoculation				

Tg: transgenic, α-Syn: α-Synuclein, MSA: multiple system atrophy, NCI: Neuronal cytoplasmic inclusion, NNI: Neuronal nuclear inclusion, GCI: Glial cytoplasmic inclusion, GNI: Glial neuclear inclusion.

## Data Availability

No new data were created or analysed in this study. Figure 1 demonstrates the established concept of pathological hallmarks of MSA [19]. Figure 3 was generated from the case of preclinical MSA, reported by Kon *et al.* [71]. Data sharing is not applicable to this article.

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
