# Peer review of "Cells2025, 14(24), 1966;https://doi.org/10.3390/cells14241966"

_cells, 2025, doi:10.3390/cells14241966_

Round 1

Reviewer 1 Report

Comments and Suggestions for Authors

The manuscript entitled "Pathological and molecular insights into the early stage of multiple system atrophy" would be attractive to readers. However some points are suggested for improving it.

  • Recently described clinical findings in the early stage of MSA are lacking. The potential of identifying some early manifestations could enrich the section 2 of your manuscript.
  • Are some studies describing relationship (plots) between the pathological studies and clinical findings? some other descriptor for cases with minimal changes-MSA?
  • How the in vitro or in vivo models of MSA could be considered as adequate? How you (or other authors) can evaluate an adequate representation of that disease? Is all in function of relationship of pathological findings? 
  • I suggest a detailed description of authors' contribution.
  • The most of references are relevant. However, the abundance of recent papers is desirable (as your manuscript is described as an update).

Author Response

Reviewer 1

The manuscript entitled "Pathological and molecular insights into the early stage of multiple system atrophy" would be attractive to readers. However some points are suggested for improving it.

  • Recently described clinical findings in the early stage of MSA are lacking. The potential of identifying some early manifestations could enrich the section 2 of your manuscript.
  • Are some studies describing relationship (plots) between the pathological studies and clinical findings? some other descriptor for cases with minimal changes-MSA?
  • How the in vitro or in vivo models of MSA could be considered as adequate? How you (or other authors) can evaluate an adequate representation of that disease? Is all in function of relationship of pathological findings? 
  • I suggest a detailed description of authors' contribution.
  • The most of references are relevant. However, the abundance of recent papers is desirable (as your manuscript is described as an update).

  1. Recently described clinical findings in the early stage of MSA are lacking. The potential of identifying some early manifestations could enrich the section 2 of your manuscript.

Response

We thank Reviewer 1 for the constructive comments. In line with your kind suggestion, we have added a new Section 4 describing the clinical features of early-stage MSA (page 6, lines 18–31 and page 7, lines 1–22) and have cited eight relevant reports (references 60–67). In addition, in this revision we have revised Section 5.2 (Section 4.2 in the original version) on page 8, line 27.

Manuscript, the section 4, (page 6, line 1831; page 7, line 122)

  1. Clinical features in the early stage of MSA

In the current consensus diagnostic criteria for MSA, a category of prodromal MSA has been proposed to capture patients in the early stage of the disease. This category comprises three layers: (i) clinical non-motor features, (ii) motor features, and (iii) the absence of exclusion criteria [1]. Among these, non-motor features serve as the entry criterion and include rapid eye movement sleep behaviour disorder (RBD), neurogenic orthostatic hypotension, and urogenital failure. Indeed, approximately 50% of patients with MSA present initially with autonomic symptoms [60], such as cardiovascular autonomic failure, urogenital and sexual dysfunction, and orthostatic hypotension [61]. In addition, RBD may predate the onset of motor or autonomic symptoms in Lewy body diseases (Parkinson’s disease and dementia with Lewy bodies) and in MSA by months or even years. Jung et al. assessed the frequency of 13 motor and non-motor signs or symptoms—some regarded as “red flags” for MSA—in 61 patients within three years of disease onset. Dysarthria (98.4%) was the most frequent feature, followed by sexual dysfunction (95.1%), probable RBD (90.2%), constipation (82.0%), snoring (70.5%), dysphagia (68.9%), and stridor (42.6%), in addition to parkinsonism and cerebellar ataxia [62]. These observations support the concept that early-stage MSA is characterised by diverse and often prominent non-motor manifestations. However, neurodegeneration in MSA variably involves the striatonigral, olivopontocerebellar, and autonomic systems, and some patients show involvement of only one or two systems in the early stage. In fact, 9% of patients with MSA do not exhibit autonomic dysfunction at any point during the disease course [63]. Furthermore, other parkinsonian disorders, including Parkinson's disease and progressive supranuclear palsy, can clinically masquerade as MSA during life, as these patients can also exhibit parkinsonism with severe autonomic dysfunction [64]. Conversely, up to half of patients with early-stage MSA show a favourable response to L-DOPA, reflecting relatively mild neurodegeneration in the putamen [3, 6]. Distinguishing such cases from Parkinson’s disease is therefore difficult, particularly in the absence of cerebellar ataxia and/or marked autonomic failure [65]. Compounding this challenge, RBD is not specific to MSA or Lewy body diseases: it can also be caused by progressive supranuclear palsy, narcolepsy, the use of antidepressants, alcohol abuse, and various autoimmune diseases, including limbic encephalitis, anti-IgLON5 antibody–associated disease, multiple sclerosis, and Guillain–Barré syndrome [66, 67]. Taken together, these factors underscore that an accurate clinical diagnosis of MSA at an early stage remains challenging when based solely on the constellation of clinical signs and symptoms. The development of reliable biomarkers for early-stage MSA is an urgent, unmet need.

Manuscript, Reference, (reference 6067)

  1. Wenning, G.K.; Scherfler, C.; Granata, R.; Bösch, S.; Verny, M.; Chaudhuri, K.R.; Jellinger, K.; Poewe, W.; Litvan, I. Time course of symptomatic orthostatic hypotension and urinary incontinence in patients with postmortem confirmed parkinsonian syndromes: a clinicopathological study. J Neurol Neurosurg Psychiatry 1999, 67, 620-623, doi:10.1136/jnnp.67.5.620.
  2. Chelban, V.; Catereniuc, D.; Aftene, D.; Gasnas, A.; Vichayanrat, E.; Iodice, V.; Groppa, S.; Houlden, H. An update on MSA: premotor and non-motor features open a window of opportunities for early diagnosis and intervention. J Neurol 2020, 267, 2754-2770, doi:10.1007/s00415-020-09881-6.
  3. Jung, Y.J.; Kim, H.J.; Yoo, D.; Choi, J.H.; Im, J.H.; Yang, H.J.; Jeon, B. Various Motor and Non-Motor Symptoms in Early Multiple System Atrophy. Neurodegener Dis 2019, 19, 238-243, doi:10.1159/000507292.
  4. Wilkens, I.; Bebermeier, S.; Heine, J.; Ruf, V.C.; Compta, Y.; Molina Porcel, L.; Troakes, C.; Vamanu, A.; Downes, S.; Irwin, D.J.; et al. Multiple System Atrophy Without Dysautonomia: An Autopsy-Confirmed Study. Neurology 2025, 105, e214316, doi:10.1212/WNL.0000000000214316.
  5. Miki, Y.; Foti, S.C.; Asi, Y.T.; Tsushima, E.; Quinn, N.; Ling, H.; Holton, J.L. Improving diagnostic accuracy of multiple system atrophy: a clinicopathological study. Brain 2019, 142, 2813-2827, doi:10.1093/brain/awz189.
  6. Miki, Y.; Tsushima, E.; Foti, S.C.; Strand, K.M.; Asi, Y.T.; Yamamoto, A.K.; Bettencourt, C.; Oliveira, M.C.B.; De Pablo-Fernandez, E.; Jaunmuktane, Z.; et al. Identification of multiple system atrophy mimicking Parkinson's disease or progressive supranuclear palsy. Brain 2021, 144, 1138-1151, doi:10.1093/brain/awab017.
  7. Barone, D.A. Secondary RBD: Not just neurodegeneration. Sleep Med Rev 2024, 76, 101938, doi:10.1016/j.smrv.2024.101938.
  8. Baldelli, L.; Calandra-Buonaura, G. Shedding light in REM sleep behavior disorder in progressive supranuclear palsy: window into neurodegeneration or diagnostic challenge? Neurology 2025, 104, e213449, 10.1212/WNL.0000000000213449.

Manuscript, the section 5.2, (page 8, line 27)

The term “early-stage” MSA has not been clearly defined. In this review, following previous reports [75-77], early-stage MSA was defined as the period within 3 years of disease onset. At this stage, varying degrees of neurodegeneration can be observed in the striatonigral, olivopontocerebellar, and autonomic systems. To elucidate the molecular alterations in early-stage MSA, analyses of proteins [78-80], cytokines [81], microRNAs (miRNAs) [82, 83], and metabolites [84] in the cerebrospinal fluid (CSF) or plasma obtained from patients within 3 years of onset have been performed.

  1. Are some studies describing relationship (plots) between the pathological studies and clinical findings?

Response

In the original version of the manuscript, we described the key relationships between motor symptoms and pathological findings on page 4, lines 15–18, and in Fig. 2. However, in accordance with your suggestion, in this revision we have additionally included the relationships between cognitive impairment and the number of neuronal cytoplasmic inclusions (page 4, line 24), as well as between L-DOPA–responsive motor dysfunction and neurodegeneration in the putamen (page 7, lines 11, 12).

Manuscript, the section 2, (page 4, line 1518)

Early in the disease course, either the striatonigral or the olivopontocerebellar system is preferentially affected; however, with disease progression, both systems become severely degenerated, often to a comparable degree (Fig. 2) [20, 21].

Manuscript, the section 2, (page 4, line 24)

Importantly, the variable distribution and extent of α-Syn-positive inclusions (GCIs, GNIs, NCIs, and NNIs) and neuronal loss likely underlie the broad clinical heterogeneity of MSA. Indeed, we previously reported the accumulation of α-Syn oligomers and subsequently increased number of NCIs in the hippocampus as a pathological substrate of cognitive impairment in MSA [11, 15, 22].

Manuscript, the section 2, (page 7, line 11, 12)

Conversely, up to half of patients with early-stage MSA show a favourable response to L-DOPA, reflecting relatively mild neurodegeneration in the putamen [3, 6].

  1. some other descriptor for cases with minimal changes-MSA?

Response

In the original version of the manuscript, we summarised all studies on “minimal change” MSA. In this revision, we have further elaborated on the pathological features in Section 5.3 (Section 4.3 in the original version).

Manuscript, the 5.3 section, (page 10, line 12, 13)

Typically, in MSA, varying degrees of neuronal loss are observed within the striatonigral and olivopontocerebellar systems [94, 95], accompanied by the widespread occurrence of GCIs and NCIs throughout the brain [96, 97]. In contrast, “minimal change” MSA represents an atypical pathological variant characterised by neuronal loss restricted to specific regions with an otherwise typical distribution of GCIs [94, 98-102]. To date, nine autopsy cases with minimal changes in MSA have been reported [98-102]. The initially described affected regions were the substantia nigra and locus coeruleus [94, 99, 101, 102]. Compared with typical MSA, patients with “minimal change” MSA showed a greater burden of NCIs, but not GCIs, in the caudate and substantia nigra [99]. However, subsequent reports documented cases in which neuronal loss was restricted to the olivopontocerebellar system [100] or limbic system [98].

  1. How the in vitro or in vivo models of MSA could be considered as adequate? How you (or other authors) can evaluate an adequate representation of that disease? Is all in function of relationship of pathological findings? 

Response

Thank you for your incisive comment. We fully agree that the adequacy of in vitro and in vivo models in the present review must be carefully considered. These models cannot fully capture the complexity of human disease dynamics, including onset timing and species-specific factors, and such shortcomings represent inevitable limitations of current experimental approaches. Nevertheless, in this review we describe the use of human induced pluripotent stem cell–derived models to approximate human pathogenesis as closely as possible, and in vivo models that are designed to recapitulate key pathological features of MSA.

In this revision, we have explicitly highlighted these inevitable yet important limitations of both in vitro and in vivo models (page 17, lines 11–15).

Manuscript, Conclusion (page 17, line 1115)

In the present review, we comprehensively summarised the current understanding of early-stage MSA. Notably, several shared features, such as mitochondrial dysfunction and specific neuroinflammation, consistently emerged in both the in vitro and in vivo MSA models. However, critical questions remain regarding whether these features reflect genuine pathogenic processes or are influenced by methodological differences and species-specific factors. In fact, both in vitro and in vivo models are designed to replicate some of the key features of early-stage MSA, based on the hypothesised sources and propagation pathways of abnormal α-Syn. These models do not fully replicate the complexity of human disease dynamics, such as onset timing and species-specific factors. Further studies using postmortem brains from patients with preclinical or early-stage MSA are warranted to clarify the upstream mechanisms driving the pathogenesis of MSA.

  1. I suggest a detailed description of authors' contribution.

Response

We have detailed author’s contribution (page 17, line 20–23).

Manuscript, Author contribution (page 17, line 2023)

YM conceived the concept of the manuscript. MTM and YM wrote the manuscript. MTM, YM, TK, FM and KW determined GCI localisations, MSA stages in Fig. 2, and prepared table 1−4. MTM and YM prepared Fig. 2. TK, FM and KW advised 2−6.2 sections in the manuscript.  All authors read and approved the final version of the manuscript and figures.

  1. The most of references are relevant. However, the abundance of recent papers is desirable (as your manuscript is described as an update).

Response

In this review, we cited milestone studies. However, in accordance with your suggestion, we have added recent papers especially about the clinical features in early-stage MSA (reference 60–67).

Manuscript, Reference, (reference 6067)

Please see described above.

Reviewer 2 Report

Comments and Suggestions for Authors

The review titled “Pathological and Molecular Insights into the Early Stage of Multiple System Atrophy” provides a comprehensive overview of current knowledge regarding early pathological and molecular alterations in Multiple System Atrophy (MSA), a rare neurodegenerative disorder marked by degeneration of the striatonigral and olivopontocerebellar systems.

The manuscript encompasses sections on clinical characteristics, early-stage pathology, molecular mechanisms, and both human studies and experimental models (in vivo and in vitro). Additionally, it addresses biomarker research, highlighting cerebrospinal fluid and plasma protein profiles as well as microRNA changes with potential diagnostic relevance.

The review is well-structured and offers a thorough and coherent discussion of the topic.

The main limitations of the review include the small number of reported preclinical autopsy cases and the intrinsic constraints of induced pluripotent stem cell (iPS) and animal models, which do not fully replicate the complexity of human disease dynamics, such as onset timing and species-specific factors.

I have just one critical observation regarding Table 2. Its improper formatting makes it difficult to interpret accurately. In addition, the font size is excessively small, which significantly impairs readability.

Author Response

Reviewer 2

The review titled “Pathological and Molecular Insights into the Early Stage of Multiple System Atrophy” provides a comprehensive overview of current knowledge regarding early pathological and molecular alterations in Multiple System Atrophy (MSA), a rare neurodegenerative disorder marked by degeneration of the striatonigral and olivopontocerebellar systems.

The manuscript encompasses sections on clinical characteristics, early-stage pathology, molecular mechanisms, and both human studies and experimental models (in vivo and in vitro). Additionally, it addresses biomarker research, highlighting cerebrospinal fluid and plasma protein profiles as well as microRNA changes with potential diagnostic relevance.

The review is well-structured and offers a thorough and coherent discussion of the topic.

The main limitations of the review include the small number of reported preclinical autopsy cases and the intrinsic constraints of induced pluripotent stem cell (iPS) and animal models, which do not fully replicate the complexity of human disease dynamics, such as onset timing and species-specific factors.

I have just one critical observation regarding Table 2. Its improper formatting makes it difficult to interpret accurately. In addition, the font size is excessively small, which significantly impairs readability.

  1. The main limitations of the review include the small number of reported preclinical autopsy cases and the intrinsic constraints of induced pluripotent stem cell (iPS) and animal models, which do not fully replicate the complexity of human disease dynamics, such as onset timing and species-specific factors.

Response

We would like to thank Reviewer 2 for your constructive comments. We acknowledge that in vitro and in vivo models of MSA do not fully recapitulate the complexity of the human disease; however, this represents an inevitable limitation of current scientific approaches. In this review, we have highlighted these important limitations in the Conclusion section (page 17, lines 11–15).

Manuscript, Conclusion (page 17, line 1115)

In the present review, we comprehensively summarised the current understanding of early-stage MSA. Notably, several shared features, such as mitochondrial dysfunction and specific neuroinflammation, consistently emerged in both the in vitro and in vivo MSA models. However, critical questions remain regarding whether these features reflect genuine pathogenic processes or are influenced by methodological differences and species-specific factors. In fact, both in vitro and in vivo models are designed to replicate some of the key features of early-stage MSA, based on the hypothesised sources and propagation pathways of abnormal α-Syn. These models do not fully replicate the complexity of human disease dynamics, such as onset timing and species-specific factors. Further studies using postmortem brains from patients with preclinical or early-stage MSA are warranted to clarify the upstream mechanisms driving the pathogenesis of MSA.

  1. I have just one critical observation regarding Table 2. Its improper formatting makes it difficult to interpret accurately. In addition, the font size is excessively small, which significantly impairs readability.

Response

For improving Table 2 readability, we separated Table 2 into new three Tables: in vitro models, genetically modified or AAV-injected animal models, and inoculation models. Accordingly, we have modified the Section 6.1.4, 6.2.2 and 6.2.3 (originally Section 5.1.4, 5.2.2 and 5.2.3) (page 13, line 19; page 15, line 30, 31; page 16, line 24)

Table 2

Table 3

Table 4

Manuscript, Section 6.1.4 (page 13, line 19)

Regardless of the cell type examined, morphological and functional abnormalities of the mitochondria appear to be common features of early-stage MSA. All in vitro models of MSA are summerised in Table 2.

Manuscript, Section 6.2.2 (page 15, line 30, 31)

Similar to the rhesus macaque model, microglia were activated in the putamen along with T cell infiltration. All genetically modified and AAV animal models of MSA are summarised in Table 3.

Manuscript, Section 6.2.3 (page 16, line 24)

The 1B fibrils closed thioflavin T-binding pockets [47], suggesting that such closed pockets may directly contribute to the formation of MSA-like pathology. All in vivo inoculation models of MSA are summerised in Table 4.

Reviewer 3 Report

Comments and Suggestions for Authors

The manuscript is of interest, there is a great need for disease-modifying therapies for neurodegenerative disorders. To achieve this, better understanding of the underlying mechanism is needed especially at the early stage of the disease. The manuscript is well-structured and informative.

Author Response

Reviewer 3

The manuscript is of interest, there is a great need for disease-modifying therapies for neurodegenerative disorders. To achieve this, better understanding of the underlying mechanism is needed especially at the early stage of the disease. The manuscript is well-structured and informative.

Thank you for Reviewer 3 reviewing our manuscript and giving positive comments.
